# Community risk perception and barriers for the practice of COVID-19 prevention measures in Northwest Ethiopia: A qualitative study

Aragaw Tesfaw[1]*, Getachew Arage[2], Fentaw Teshome[1], Wubet Taklual[1], Tigist Seid[3], Emaway Belay[1], Gashaw Mehiret[4]

1 Department of Public Health, College of Health Sciences, Debre Tabor University, Debre Tabor, Ethiopia,
2 Department of Nursing, College of Health Sciences, Debre Tabor University, Debre Tabor, Ethiopia,
3 Department of Midwifery, College of Health Sciences, Debre Tabor University, Debre Tabor, Ethiopia,
4 School of Medicine, College of Health Sciences, Debre Tabor University, Debre Tabor, Ethiopia

* aragetesfa05@gmail.com

**Data Availability Statement:** All relevant data are within the paper and its Supporting Information files.

## Abstract

### Background

According to the World Health Organization, viral diseases continue to emerge and represent a serious issue for public health. The elderly and those with underlying chronic diseases are more likely to become severe cases. Our study sets out to present in-depth exploration and analyses of the community's risk perception and barriers to the practice of COVID-19 prevention measures in South Gondar Zone, Northwest Ethiopia.

### Methods

A qualitative study was done in three districts of South Gondar Zone. Community key informants and health extension workers were selected purposely for in-depth interviews and focus group discussion. The interviews were conducted by maintaining WHO recommendations for social distancing and use of appropriate personal protective equipment. The sample size for the study depended on the theoretical saturation of the data at the time of data collection. The qualitative data generated from in-depth interviews and focus group discussions was transcribed verbatim and translated into English language and thematically analyzed using open code software version 4.02.

### Results

Three main themes and five categories emerged from the narrations of the participants regarding the perceived barriers for the practice of COVID-19 prevention measures. A total of 9 community key informants (5 women development armies (HDA), 2 health extension workers (HEW), and 2 religious leaders participated in the in-depth interview, while two focus group discussions (7 participants in each round) were conducted among purposely selected community members. The age of the participants ranged from 24 to 70 years with

**Funding:** The author(s) received no specific funding for this work.

**Competing interests:** The authors declare that they have no competing interests.

the median age of 48 years. The major identified barriers for practicing COVID-19 prevention measures were the presence of strong cultural and religious practices, perceiving that the disease does not affect the young, misinformation about the disease, and lack of trust in the prevention measures.

## Conclusions

Socio-cultural, religious, and economic related barriers were identified from the participant's narratives for the practice of COVID-19 prevention measures in south Gondar Zone. Our findings suggest the need to strengthen community awareness and education programs about the prevention measures of COVID-19 and increase diagnostic facilities with strong community-based surveillance to control the transmission of the pandemic.

## Background

The world is closely watching the outbreak of respiratory illness associated with the novel beta coronavirus SARS-CoV-2 [1]. Initially termed 2019-nCoV, sequencing showed that the now officially named SARS-CoV-2 is 80–89% similar to bat severe acute respiratory syndrome-related coronaviruses found in Chinese horseshoe bats [2]. Prior to the global outbreak of SARS-CoV in 2003, HCoV-229E and HCoV-OC43 were the only coronaviruses known to infect humans. Following the SARS outbreak, 5 additional coronaviruses have been discovered in humans, most recently the novel coronavirus COVID-19, believed to have originated in Wuhan, Hubei Province, China. SARS-CoV and MERS CoV are particularly pathogenic in humans and are associated with high mortality [2, 3].

The world is now suffering from this global pandemic which has killed millions of people worldwide [4, 5]. As evidence shows, all people in the world are susceptible to this viral respiratory tract disease, although the impact is different depending on several socio-demographic, economic, and other infrastructural differences. There are no any proven pharmacological treatments developed for the disease until now, although a number of trials are currently underway by scientists across the globe. However, COVID-19 vaccines are developed within the shortest period in the history of vaccine production and now become available. Ethiopia also received AstraZeneca vaccines manufactured by Serum Institute of India (SII) on 6 March 2021 [6, 7]. However, there are scientifically accepted and recommended prevention measures against COVID-19 infection such as social distancing, frequent hand washing, use of face mask, and proper use of alcohol-based hand rubs or sanitizers [8–10].

Ethiopia is at high risk for the introduction and spread of the novel coronavirus disease (COVID-19) and the effect might be devastating because of the multiple health challenges the country already faces: rapid population growth and increased movement of people; existing endemic diseases, such as human immunodeficiency virus, tuberculosis, malaria and increasing incidence of non-communicable diseases [11]. Although some parts of the world are returning to pre COVID levels of engagement, the risk of the pandemic is continued and becomes a global challenge [12]. According to the September 2, 2021 Worldometer report, currently India and South Africa became the number one affected countries from Asian and African countries respectively [13]. Prevention practices are critical to combat the spread of Coronaviruses (CoVs) [9]. Poor prevention practices can lead to economic, social, and political crisis, and increased risk of death. The benefits of COVID-19 prevention, particularly in the developing setting like Ethiopia are invaluable because of the deprived health system [11, 14, 15].

To contain the pandemic, the country established a National Ministerial Committee and the government declared a state of emergency on April 8, 2020. On March 16, 2020, the committee released guidelines that included: 1) "A 14 days quarantine of passengers (travelers) in isolated centers; 2) Ethiopian Airlines to stop flying to 30 countries: 3) regulating the market to avoid unethical exploitation of the situation:4)stopping religious gatherings; 5)avoiding public overcrowding; 6) strictly refraining from harassing foreign nationals; 7) strict adherence to self- prevention and protection; 8) medical professional training;9) facilitating the acquisition of testing facilities; 10) temporarily closing of night clubs and bars; 11) regulating market to avoid unethical exploitation of the situation; and 12) supporting regions preparedness to contain the disease" [8, 11, 15].

In this study, healthcare workers and key informants in the community including local leaders were interviewed either individually or as part of focus groups to share their perceived barriers to COVID-19 prevention measures. This is helpful from the view of the following perspectives. Healthcare workers are the frontline in battling the spread infection as a whole [16]. Secondly, community leaders have a great role in the prevention of COVID-19 infection in the community [17].

The basic protective measures against COVID-19 including distancing, personal protective equipment, and handwashing as noted above are poorly practiced in the community of South Gondar Zone, along with the above scientific explanations, controlling the spread of infection is essential to preventing outbreaks of COVID-19 [18]. Basic protective measures against the new coronavirus include staying at home, washing hands with soap and water, frequently, using alcohol-based hand rub or washes, use of personal protective equipment, and keeping physical/social distance [9, 15, 19]. Thus, the implementation of these practices to reduce the spread of COVID-19 would be challenging in Ethiopia. Consequently, the perception of health extension workers and community leaders as a key informant are important to underpin the prevention of COVID-19 Infection [11, 20]. In response, the present qualitative study was designed to explore the community risk perception about COVID-19 and the perceived barriers to practice recommended prevention measures among community key informants and other community member's in South Gondar Zone, North west Ethiopia.

## Methods and materials

### Study area and period

This study was conducted on purposely selected Woredas (the lowest administrative areas called woreda's or districts in Ethiopia, government structure which form zone [21] of South Gondar Administrative Zone (Worata, Libokemkem (Adiszemen) and Guna Begemidir woreda (Kimirdingay) from May 15 to June 01, 2020. There are eight primary and one comprehensive specialized hospital and more than 187 health centers in the zone. According to the 2007 E.C population census report, the total population of the zone is around 2,578,906. Currently, there is one isolation center prepared at Atse Seyife Yared Health center and one quaratine center at Debre Tabour University. There is also one treatment center at the zonal level. However, COVID-19 treatment centers are not still established at the woreda level.

### Study design

A qualitative study using a phenomenological approach was used to explore the perceptions, thinking, feelings, and experience of community members and health extension workers on a specific topic to explore the community risk perceptions and barriers related to the practice of COVID-19 prevention measures in the community. This study follows COREQ guidelines for reporting qualitative studies [22].

## Inclusion criteria

✓ People resided for at least six months in the area

✓ People with age greater than 18 years

✓ Community key informants (health extension workers, Women, health development armies, and religious leaders)

✓ Health extension workers who have been working more than two years in the respective sites

## Participants of the study and sampling procedures

The participants of this qualitative study were purposely selected community members and health extension workers. The community key informants (Health extension workers, women health development armies, and religious leaders) were selected for in-depth interviews with the assumption of having detail information about the issue in each community, while other community members were selected for focus group discussions in two rounds (7 participants in each round) at the selected woreda's. We determined the sample size based on a theoretical saturation point in data collection time when new data no longer bring additional information to the research question. Therefore, the interview process was continued until that point was obtained. Finally, a total of 9 community key informants for in-depth interview and 14 individuals from the community were participated in focus group discussions.

## Data collection tools and procedures

In-depth interviews and focus group discussions were conducted using a semistructured in-depth interview and focus group discussion guide to facilitate the interviews (see S2 and S3 Files). The guide was developed by the research team by reviewing similar literatures on the area. It was first developed in English and then translated to the local language (Amharic) to facilitate communication with participants.

The guides were focused on the community's experiences with the practice of the recommended prevention methods of COVID-19, their risk perceptions on acquiring the disease, and the barriers to practice it in the community. All interviews and discussions were tape-recorded and written notes were also taken during the interviews. Participants of the study were selected with the help of the health local leaders. After we obtained informed consent to participate, an appropriate place and time for an interview and focus group discussion was arranged in a private setting. Two public health professionals who had a Master of public Health in health education and Epidemiology collected the data with the assistance of two note takers. Interviewers and facilitators were fluent in written and spoken Amharic and English language. All interviews were conducted within the community setting and lasted 30–60 minutes.

The transferability of the findings was established by collecting data on randomly selected woreda's in the administrative zone with community key informants who are representatives of the community and having detail information about the people in their surroundings. This helped to get more information from different perceptions of the community with different socio-demographic and cultural experiences. To maintain the validity of the findings, the investigators developed a rapport with participants who were participated in the study. Credibility was maintained through participant checking during in-depth interviews and focus group discussions, and through feedback of findings at the end of the study from whom the

data was taken. Keeping a record with information about imitation was enhanced conformability. The dependability of data was maintained by taking the depth of information until reaching the data saturation point.

## Data analysis procedure

The qualitative data collected from in-depth interviews and focus group discussions were transcribed verbatim and translated into English language and thematically analyzed using open code software version 4.02 by relistening the tape recorder several times and reading the field notes line by line. The transcripts were carefully read and entered in to open code software. Transcripts and translations were cross-checked for accuracy and consistency by two independent persons. First, repeated responses for each question were identified, and then similar responses and grouped into codes. The coding process was continued until the data was exhausted. Then, the codes were categorized and themes were merged.

## Data quality assurance

To maintain data quality, the supervisor and data collectors were trained for two days on the basic principles of data collection, study overview, and how to do other related activities during data collection by the principal investigator. Strict daily supervision of the data collection process were maintained throughout the data collection period. One supervisor was responsible for one study site. The data collectors were debriefed each day after the data collection by the supervisors.

## Ethical approval and consent to participate

Ethical approval letter was obtained from Debre Tabor University. Official letter of co-operation was written to community leaders and health extension workers to obtain their co-operation in facilitating the study. Oral informed consent was obtained from each participant prior to data collection after information on the study was explained to them (see S1 File). This is because the study is just a qualitative research which did not have any intervention /clinical trial and did not have follow-ups. In addition, some of the participants were illiterate, so taking oral consent was sufficient after giving a detail explanation on the aim of the study to the participants (see the details of the study information sheet in S1 File). The ethical committee was approved the oral consent procedure. Confidentiality of information was assured by excluding names and identification in the interviews. In-depth interviews and focus group discussions were conducted by maintaining WHO recommendations for social distancing and use of appropriate personal protective equipment's.

## Results

### Socio-demographic characteristics of study participants

The table below describes the socio-demographic characteristics of focus group discussions and in-depth interview participants for this particular qualitative study. A total of 9 in-depth interviews and 2 focus group discussions were conducted. The age of the participants ranged from 24 to 70 years with the median age of 46 years. The majority 14 (60.9%) of the participants were females. Among the participants of in-depth interview, 2 were health extension workers, 5 were women health development armies, and 2 were religious leaders. About 19 (82.6%) were orthodox religion followers while the rest were Muslims. Regarding marital status, most 14 (60.9%) of the participants were married while the rest were single. The majority 17 (73.9%) of the participants were from rural areas (Table 1).

**Table 1. Socio-demographic characteristics of study participants for in-depth interview and focus group discussion in South Gondar Zone, Northwest Ethiopia, 2020.**

| Characteristics | Frequency (n = 23) | Percentage |
|---|---|---|
| Age group | | |
| <30 | 3 | 13.0 |
| 30–39 | 5 | 21.7 |
| 40–49 | 4 | 17.4 |
| 50–59 | 5 | 21.7 |
| ≥60 | 6 | 26.1 |
| Home residence | | |
| Rural | 17 | 73.9 |
| Urban | 6 | 26.1 |
| Educational status | | |
| Illiterate | 5 | 21.7 |
| Primary education completed | 3 | 13.0 |
| Secondary education completed | 6 | 26.1 |
| College and above completed | 9 | 39.1 |
| Religion | | |
| Orthodox | 19 | 82.6 |
| Muslim | 4 | 17.4 |
| Marital status | | |
| Married | 14 | 60.9 |
| Single | 9 | 39.1 |
| Occupational status | | |
| Farmer | 9 | 39.1 |
| Government employee | 6 | 26.1 |
| House wife | 3 | 13.0 |
| Merchant | 5 | 21.7 |
| **Role in the community** | | |
| Health extension worker | 2 | 8.7 |
| Women health development army | 5 | 21.7 |
| Religious leader | 2 | 8.7 |
| Other community member* | 14 | 60.9 |

Key: * = peoples selected for focus group discussion from the community.

## Risk perception and barriers for the practice of COVID-19 prevention methods

Three main themes (Personal factors, Socio-demographic & economic related factors, cultural and religious related factors) and five categories (Lack of knowledge & awareness, Socio-demographic, economic, cultural, and religious related barriers) were emerged from the narrations of the participants regarding the risk of acquiring of COVID-19 and perceived barriers for the practice of prevention measures (see S4 File).

### Personal related barriers

**Lack of knowledge and awareness about COVID-19 infection.** As most of the participants' described, there were various reasons mentioned for not practicing COVID-19 prevention measures in the community. One of the major reasons stressed by almost all participants

for not practicing COVID-19 prevention measures was lack of knowledge and awareness about risk factors, signs, and symptoms of the disease. Most of the focus group discussion participants did not hear about the prevention measures as well as the clinical features of the disease.

> *"I heard this new disease from the health extension worker, however, I did not understand about the disease. I do not know, it will infect me or not. . ..." (A 32 -years-old female FGD participant)*

> *"I heard the name of the disease which is called "Corona'. I did not know about its characteristics. There was no such disease in our country." (A 52 -years-old female FGD participant)*

As some participants described, there is a lack of information access about COVID-19, particularly for the rural community, still some participants did not hear about the disease, while some others heard only the coming of new disease or epidemic only but they did not hear about the disease clinical features and prevention measures.

> *"I did not hear about this new disease which you called Corolla or Corona. I heard it know from you. As you know, my house is far from the health post. Unless the health extension worker comes to our Kebele, I could not hear anything about health." (A 60 -years-old women health development army)*

> *"I heard the name Corona virus disease through a telecom message in my phone. However, I did not understand about the prevention methods" (A 45 -years-old FGD participant)*

As narrated by the participants, some heard about the prevention methods, but they did not understand how to practice it. Even some participants mentioned the recommended prevention measures like hand washing with soap and water, maintaining, physical distancing, and staying at home, but none of them have detail information about how to practice them.

> *I have television and I heard the prevention methods of corona virus, but I am not aware and clear about its importance and how to practice it." (A 35 -year-old women, health development army)*

**Socio-demographic and economic related barriers.** Almost all participants explained that it is very difficult to practice all prevention measures of COVID-19, mainly it is difficult to stay at home, although it is essential to keep oneself from COVID-19 infection since almost all people need their daily consumption from their daily work. Therefore, it will cause hunger and another social chaos or distraction if it is obliged to stay at home. They explained as the stay-at-home policy could not be wrought for low-income countries such as Ethiopia.

> *"I myself could not stay at home because I could not get my daily foods because I am a daily worker." A 28 -years-old FGD participant*

> *"Ohhh. . .. How can we stay at home? For how many days? It is unthinkable, we all have work outside our homes." A 32 -years-old FGD participant*

Some of the participants relate the disease to demographic factors like age and residence. They perceived as the disease could not infect children and young's and some perceived as the disease will affect the urban population. As a result of these mentioned barriers, people fail to practice the prevention methods.

*"---I think the disease does not affect young people. Therefore, I will not be at risk of this disease. Thus I do not afraid and care about it too much"* A 24 -years-old FGD participant

*"As I heard, the disease transmission will be fast in cold environment than hot areas. We are living in a hot climatic areas, so the disease will not affect us that much." (A 40 -year-old health extension worker)*

*". . ...I am 70 years old, and I have heart problems. My child is a college student and he told me as I am at risk of this disease if it comes to our village . . ." (A 70 -years-old FGD participant)*

As the participants explained, most of them did not have adequate money and other resources for their daily lives. As they narrated, those individuals who have money will buy their essential goods for a month or more than that and they can stay at home. However, those who get their income from their daily work could not afford that and could not stay at home.

*"I think staying at home is very difficult for most people in our country since most people could not get their daily consumption, rather the impact will be worse than the effect of the disease." (A 46 -year-old Women, health development army)*

As the participants described, there is no even access to get soap for practicing hand washing mainly for the rural people. They did not get also alcohol-based hand rub/ sanitizer/.

*"I could not get adequate soap for my family to regularly wash our hands. It is costly for me and I could not afford to buy it. I do not know hand sanitizer, I only heard from you now." (A 39 -years-old FGD participant)*

Among the participants, the majority of them were from rural areas. As they said, their being in rural and remote areas makes them to not to access updated information's about the disease since they are far from the health facilities and some life in hard-to-reach areas so that their frequent contact is very low to the health care providers.

*"I live in very remote area. Unless I come to a health facility, no one can come to my village and tell me about the disease since there is not transport access to our village." (A 46 years old FGD participant).*

*"I lived in a rural area so I faced many difficulties when I came to this now because of the long distance. Since there is no road transport access." A 52 -years-old FGD participant*

The participants also explained that the absence of an educated person in the household would have a significant role for not practicing the prevention methods. Since it is difficult to understand the disease characteristics.

**Cultural and religious related barriers.** The other important barrier mentioned by the participants was considering the disease as not that much serious and even some understand as it is the disease of those who eat wild animals which are not allowed to be eaten in the bible.

*". . .As I heard from other people, corona will infect those who eat wild animals but not us." (A 38 -year-old health extension worker)*

The most frequent reason reflected by the participants for not practicing COVID-19 prevention measures was that they trusted in their religion and did not affect those who had a

strong religious practice. As they described, if we follow the right principles of religion and strongly believe on GOD, they will not be affected by the disease. All participants believe that the disease comes due to our sin and GOD wants to teach his power to the world.

> ". . .I am sure that this disease is due to our sin and the GOD wishes to teach us by this disease and he just shows his supreme power to the world as no one can do anything without the will of GOD". (A 50-year-old religious leader)

> ". . .if you strongly believe in the power of GOD, this disease will not infect you." (A 65 year's FGD old participant)

> "There were several predictions that this kind of epidemic will occur from our ancestors. Therefore, the disease is due to in human acts like homosexuality, racism." (A 55 years-old religious leader)

> "If I virus infects me, I will go to use holy water and I will pray. I will not prefer to go to the health facilities since there is no any medical treatment till know." (A 45-year-old participant, FGD said).

As the participants said, almost all participants were trusted with the use of traditional treatments prepared from different herbal plants. As the participants believed that the disease could be healed by common traditional treatment modalities. When they were asked also what they will do if they have the disease, they said as they will use these traditional treatments prepared by the local traditional healers from different plant leaves and roots and to use holy water. Most of the participants have a strong belief on the use of herbal medications and spiritual treatment options for treating corona virus than modern medical care. This all-mentioned reason may consider as barriers for practicing COVID-19 prevention measures.

> "I think the herbal medication which we used for common cold will be effective to treat corona since it has similar characteristics." (A 40-year-old participant, FGD said)

> "I think this disease is due to our sin, so praying and going to use holy water is the only option we have to do". (A 68 year-old woman, health development army)

When the participants asked about their communities' perception about the disease, there are different beliefs that made to not practice COVID-19 prevention measures. One thing that was mentioned was that their perception that the disease does not come from their locality. They believe that the government should close all boarders and should test all those who came from abroad, otherwise they think as our prevention could not have any effect on the disease.

> ". . .people in my community believe that the disease does not come from our village and they strongly recommend the government to control those who come from abroad." (A 57 -year-old FGD participant)

> "In our community we have a strong social interaction and most of the time we live together and we meet for different social gatherings like mourn, wedding, Idir, Ikub, and even we drink coffee together. In addition, we do not want to eat alone, we eat together by sharing what we have in our hands, so this is very challenging to stop in a short period of time." (A 64 -years-old FGD participant)

> "I think it is very challenging to practice prevention measures because our way of living is difficult to bring behaviour change. If someone tries to practice, some others are so careless.

*Therefore, one practice may not be enough and does not give a guarantee unless all other people surrounding you practice it."* (A 63 -year-old women, health development army)

*"As I think, most of us try to practice, but sometimes we forget it and we do our normal part of life since we do not have such culture before."* (A 50 -year-old religious leader)

Some of the participants believe that the disease could not affect people living in very hot areas. As a result, some carelessness is observed in practicing COVID-19 prevention measures.

*"As you know, we live in hot area and the people in my village believe that the disease cannot come to hot areas, the disease cannot survive in hot areas. . ."* (A 26 -years-old FGD participant)

## Discussion

As evidence shows, the proper practice of the recommended prevention methods of COVID-19 is an effective and the only measure to control the spread of the pandemic [10, 14, 18]. However, the practice of such prevention measures is not consistent from place to place and peoples usually are reluctant to practice mainly in Ethiopia, although efforts are made by the government [11, 15, 23], so we tried to explore the perceived barriers for the practice of COI-VID-19 prevention measures and community risk perception on the disease focusing on both rural and urban settings. Our study revealed that most community members perceived as they are at risk of acquiring COVID-19 infection, but lack of knowledge and awareness, socio-demographic, economic, cultural, and religious factors affect them to consistently practice prevention measures. The findings are similarly reported in other similar studies conducted in different regions of Ethiopia [17, 24–27].

Regarding social distancing, our study revealed that most people are not that much aware of practicing social distancing. As most said, they did not understand the concept of social distancing, how much distance is allowed for protection in COVID-19. People's way of living has a significant role in the implementation of COVID-19 prevention measures since most people in Ethiopia are living together in closed environments. Thus, the peoples have difficulty to maintain their social distancing. This finding is in line with a study conducted in other parts of the country in which lack of awareness is a significant contributor for improper practice of COVID-19 prevention measures [17, 25]. Lack of access to information is also a factor mentioned in our study as most people could not get updated information about the disease risk factors and prevention methods. This finding is consistent with a study conducted in Northwest Ethiopia which found the overall rate of information exposure about COVID-19 was less than 50 percent [24].

The other important recommended and effective prevention measure is stay at home; however, our findings showed that it is a very difficult preventive measure to practice since most people live in hand-to—to—mouth way of life so that they need to get their daily consumption through daily work otherwise they will face hunger. As a result of this, most people cannot stay at home. As Ethiopia is one of the low-income countries in the world with very young population and most of the people are living in the traditional way of life which is risky for contagious diseases like COVID-19. As a result of these, most people live together in a clustered environment, approximately 4–8 people in a single room. The finding is in line with other studies conducted in Ethiopia [11, 15, 28].

In our study, cultural and religious factors were mentioned as barriers for the practice of COVID-19 prevention measures. There are several strong religious and cultural practices in Ethiopia which interact with people in different social activities like mourning, wedding, Idir,

Ikub, and coffee ceremonies. Therefore, it is a challenging task for the government to change people's behaviour to practice prevention measures. Culturally, there are a number of practices which need public gathering in Ethiopia. Therefore, the presence of such strong cultural norms, myths, and beliefs affect peoples to practice COVID-19 prevention measures properly [26].

Most people also do not have access to hand washing facilities. Some are unable to get soap due to financial limitations while some others do not have access to water supply.

Our study also revealed that several religious barriers are mentioned as reasons for not practicing COVID-19 prevention measures. These barriers mentioned were strong religious believers which people spend most of their time at morning and evening at churches and mosques which make them to expose to each other. Peoples are reluctant to stop to go to public gathering areas like churches and mosques. There is a strong belief in the community that this disease could only be tackled through keeping religious activities since most of the society believe that the disease is caused due to people's sin as GOD's punishment. The findings are similar to other study findings in Ethiopia in which strong religious practices make people not to trust on prevention methods, rather prefer to pray together and strongly engaging in other religious activities to combat the pandemic [26, 28, 29]. This perception may also become challenges in the acceptance of COVID-19 vaccine if supplies become available in the area unless community sensitization to dispel myths and misconceptions. Still, the acceptance of the vaccine is low in Ethiopia (31.4%) [30].

Our study implies that, although community members perceived as they will be at risk of acquiring the disease, their practice of recommended prevention methods is nil, which is also a national problem encountered in the country [11, 15]. Similarly as a study in Gondar city found, the overall prevalence of good adherence towards COVID-19 prevention measures was 51.04% [28].

## Strength of the study

Based on the researcher knowledge, this study is the first qualitative study in the area which explored the risk perceptions of the community and the perceived barriers to the practice of COVID-19 prevention measures from community key informants and other community member's perspective using in-depth interviews and focus group discussions.

**Limitations of the study.** Since the majority of the participants were from a rural areas, the residential status may be a determinant of the perceived barriers.

## Conclusion

Lack of knowledge and awareness about the disease, socio-demographic barriers, presence of strong cultural and religious practices, perceiving that the disease does not affect the young, misinformation about the disease, and lack of trust on the prevention measures were the major perceived barriers for the practice of COVID-19 prevention measures explored in this study. Our findings show the need for creating community awareness and education programs about the prevention measures of COVID-19, and it is essential to increase diagnostic facilities with strong community-based surveillance to control the transmission of the pandemic. Additionally, strict measures should be taken nationally to those who are reluctant to practice the methods to save the lives of the majority.

## Supporting information

**S1 File. Study information sheet for participants of in-depth interview and focus group discussion.**
(PDF)

**S2 File. In-depth interview guide.**
(PDF)

**S3 File. Focus group discussion guide.**
(PDF)

**S4 File. Themes and categories (subthemes) developed from the narration of in-depth interviews and focus group discussion.**
(PDF)

## Acknowledgments

Authors would like to acknowledge data collectors, supervisors, and participants for focus group discussions and in-depth interviews.

## Author Contributions

**Conceptualization:** Aragaw Tesfaw, Getachew Arage, Wubet Taklual.

**Data curation:** Aragaw Tesfaw.

**Formal analysis:** Aragaw Tesfaw.

**Investigation:** Aragaw Tesfaw.

**Methodology:** Aragaw Tesfaw, Getachew Arage.

**Software:** Aragaw Tesfaw.

**Writing – original draft:** Aragaw Tesfaw, Getachew Arage, Fentaw Teshome, Wubet Taklual, Tigist Seid.

**Writing – review & editing:** Aragaw Tesfaw, Getachew Arage, Fentaw Teshome, Wubet Taklual, Tigist Seid, Emaway Belay, Gashaw Mehiret.

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
