## [Decision Letter · Decision Letter 0]

13 Apr 2021

PONE-D-21-07870

Perceived Barriers for the Practice of Covid-19 Prevention Measures in Northwest Ethiopia: A Qualitative Study

PLOS ONE

Dear Dr. Tesfaw,

Thank you for submitting your manuscript to PLOS ONE. After careful consideration, we feel that it has merit but does not fully meet PLOS ONE’s publication criteria as it currently stands. Therefore, we invite you to submit a revised version of the manuscript that addresses the points raised during the review process.

In your revised version of your article, please review and respond to each comment from the reviewers. We look forward to receiving your revised version.

We look forward to receiving your revised manuscript.

Kind regards,

Amy Michelle DeBaets, PhD

Academic Editor

PLOS ONE

Journal Requirements:

3. Please specify in your ethics statement:

a) whether the ethics committee approved the verbal/oral consent procedure,

b) why written consent could not be obtained, and

c) how verbal/oral consent was recorded.

4. When reporting the results of qualitative research, we suggest consulting the COREQ guidelines: http://intqhc.oxfordjournals.org/content/19/6/349. In this case, please consider including more information on the number of interviewers, their training and characteristics; and please provide the interview guide used.

5. Please include a copy of Table 2 which you refer to in the Results section of your text.

Reviewers' comments:

Reviewer's Responses to Questions

**Comments to the Author**

1. Is the manuscript technically sound, and do the data support the conclusions?

Reviewer #1: Partly

Reviewer #2: Yes

2. Has the statistical analysis been performed appropriately and rigorously? 

Reviewer #1: N/A

Reviewer #2: Yes

3. Have the authors made all data underlying the findings in their manuscript fully available?

Reviewer #1: No

Reviewer #2: Yes

4. Is the manuscript presented in an intelligible fashion and written in standard English?

Reviewer #1: No

Reviewer #2: Yes

5. Review Comments to the Author

Reviewer #1: The quotes presented in this paper are very valuable, however, given that they are from nearly a year ago it would be useful to place them into context with the state of the pandemic now. Have the cases become unmanageable, likely as a result of the identified barriers, or was the government about to step up and provide a widespread educational campaign/provide mask, etc.? The paper, in particular the methods section, is hard to follow. The journal is likely able to help with grammar, but there are also several typos and the methods section is repetitive without making the recruitment strategy clear.

For example:

1. For the “source and study population” section, it is noted that “all people residing in South Gondar zone…were the source populations…” But I imagine individuals recruited for the in-depth interviews were patients in the clinic for the day? Either way the specific way they were targeted/recruited needs to be specified.

2. Similarly, how were the members for the focus group recruited? While the sample size for qualitative studies is generally low, those in this study cover a large geographic region, large spread in age, and professions. For this reason, it would be useful to provide some kind of indication of who is saying what quote and potentially a table of frequencies per theme/category…maybe also separate this by community member vs. key informants.

3. Please provide the interview guide

4. Why was the interview guide created in English? Are there subtleties in the native language that would provide different meaning?

Reviewer #2: Abstract: the last sentence of results is incomplete as same as the first one of conclusion.

Abstract/.conclusion: don’t repeat results as conclusion

Introduction: a short looking at current literature/previous studies improves the introduction.

Methods: don’t use future verbs

Methods: sampling method needs more exploration, it is to somewhat unclear in the current form.

Methods: as the majority of participants were from rural regions, it should be notes as a limitation considering that the residential status may be a determinant of perceived barriers.

Results: inserting a table including the main themes and sub-themes in the beginning of results is suggested.

Discussion: the first 2 paragraphs of discussion should e removed into introduction section. Start your discussion with brief presentation of main results.

Discussion: discussion section should be improved by providing some policy recommendation/applications of results to deal with the problem and improve the status.

6. PLOS authors have the option to publish the peer review history of their article (what does this mean?). If published, this will include your full peer review and any attached files.

Reviewer #1: No

Reviewer #2: **Yes: **Mohammad Amin Bahrami

---

## [Author Response · Author response to Decision Letter 0]

18 May 2021

Dear, Editor,

Greetings!

Thank you very much for all the comments provided regarding our manuscript entitled “Perceived Barriers for the Practice of Covid-19 Prevention Measures in Northwest Ethiopia: A Qualitative Study” which are fully accepted and included in the revised version. I have accordingly made necessary revisions on the paper following the comments provided from the reviewers and editor. I also attached the themes and categories (Sub-themes) as supporting information file 4 and in-depth interview and Focus group discussion guides as supporting information file 2&3 for the reviewers based on their request. in addition I attached the study information sheet and consent form as supporting information file-1. For your kind consideration, please find a point by point response to the comments in the next page of this letter and a submitted new revised version of the manuscript. 

All new changes have been highlighted in dark blue in the main document in order to facilitate review.

I hope that you will find the edits as per your expectation and I look forwards to hear from you

soon.

Yours Sincerely,

 Aragaw Tesfaw (MPH, Epidemiology)

Lecturer, Department of Public health 

College of health sciences, Debre Tabor University 

Email: aragetesfa05@gmail.com

Phone: 251921743820

I. Point by point responses to editor comments 

- Author Responses: corrected /edited based on the given editor comment and we prepared the manuscript based on the PLOS ONE author guide line. 

- Author Reponses: We invited English language editor for grammar edition and she corrected grammatical issues accordingly based on the recommendation. in addition we used on line grammar editing services. 

Author Reponses: the name of the colleague who edit the grammar was Emaway Belay, she is health education, promotion and Behavioral science expert and she has master of public health 

Author Reponses: All new changes have been highlighted in dark blue in the main document and other requested information’s are uploaded as supporting information files 

Author Reponses: The edited manuscript is uploaded as Manuscript file

3. Please specify in your ethics statement:

a) Whether the ethics committee approved the verbal/oral consent procedure,

Author Reponses: Yes! The ethical committee was approved the oral consent procedure 

b) Why written consent could not be obtained, and

Author Reponses: the study is just a qualitative research which did not have any interventions /clinical trial and did not have follow ups. In addition some of the participants were illiterate so oral consent was appropriate to take in our study after giving a detail explanation on the aim of the study to the participants (see the detail of study information in supporting information file 1). 

c) How verbal/oral consent was recorded.

Author Reponses: What we’re doing is just we read the study information sheet which deals about the study objectives, the rights and confidentiality of the information obtained from participants, they were agreed to participate in the study then we included them to the study and we record their sayings (see the detail of study information in supporting information file 1) . 

4. When reporting the results of qualitative research, we suggest consulting the COREQ guidelines: http://intqhc.oxfordjournals.org/content/19/6/349. In this case, please consider including more information on the number of interviewers, their training and characteristics; and please provide the interview guide used.

 Author Reponses: 

- We used COREQ guidelines in this qualitative study and we attached the interview guide as supporting information file 2&3 

- There were two interviewers and two note takers who conducted the FGD and in-depth interviews. The interviewers are public health professionals who have master in public health specialties and had experience in qualitative research. 

5. Please include a copy of Table 2 which you refer to in the Results section of your text.

 Author Reponses: we are now put a copy of Table 2 based on the comment (see supporting information file _4) 

 Author Reponses: edited/corrected based on the comment 

II. Point by point responses to the comments for reviewers 

Comments to the Author

1. Is the manuscript technically sound, and do the data support the conclusions?

Reviewer #1: Partly

Reviewer #2: Yes

2. Has the statistical analysis been performed appropriately and rigorously? 

 Reviewer #1: N/A

 Reviewer #2: Yes

3. Have the authors made all data underlying the findings in their manuscript fully available?

Reviewer #1: No

Reviewer #2: Yes

4. Is the manuscript presented in an intelligible fashion and written in standard English?

Reviewer #1: No

Reviewer #2: Yes

 Review Comments to the Author

5. Reviewer #1: The quotes presented in this paper are very valuable, however, given that they are from nearly a year ago it would be useful to place them into context with the state of the pandemic now. Have the cases become unmanageable, likely as a result of the identified barriers, or was the government about to step up and provide a widespread educational campaign/provide mask, etc.? The paper, in particular the methods section, is hard to follow. The journal is likely able to help with grammar, but there are also several typos and the methods section is repetitive without making the recruitment strategy clear.

For example:

1. For the “source and study population” section, it is noted that “all people residing in South Gondar zone…were the source populations…” But I imagine individuals recruited for the in-depth interviews were patients in the clinic for the day? Either way the specific way they were targeted/recruited needs to be specified.

Author Reponses: individuals recruited for the in-depth interviews were not patients (it was editorial error) in the clinic for the day rather they were purposely selected community key informants (women health development armies, Health extension workers and religious leaders) for In-depth interviews while the FGD participants were selected purposely from community members with the help of local Kebele leaders. 

2. Similarly, how were the members for the focus group recruited? While the sample size for qualitative studies is generally low, those in this study cover a large geographic region, large spread in age, and professions. For this reason, it would be useful to provide some kind of indication of who is saying what quote and potentially a table of frequencies per theme/category…maybe also separate this by community member vs. key informants.

Author Reponses: 

The FGD participants were selected from the community members with the help of local leaders in the study sites. The study is conducted in one administrative zone (South Gondar zone) and it is not a regional study. The Themes, categories and the main findings /selected quotes were putted in the supporting information file based on the comment.

(See supporting information file -4) 

3. Please provide the interview guide

Author Reponses: edited/corrected as commented, we provide the interview guide as supporting information file 2&3

4. Why was the interview guide created in English? Are there subtleties in the native language that would provide different meaning?

Author Reponses: 

The guide (IDI & FGD) was first developed in English language and then translated to the local language (Amharic) to facilitate communication with participants. Interviewers and facilitators were fluent in written and spoken Amharic and English language.

6. Reviewer #2: Abstract: the last sentence of results is incomplete as same as the first one of conclusion.

Abstract/.conclusion: don’t repeat results as conclusion

Author Reponses: edited/corrected based on the comments

Introduction: a short looking at current literature/previous studies improves the introduction.

Author Reponses: edited/corrected as commented

Methods: don’t use future verbs

Author Reponses: edited/corrected as commented

Methods: sampling method needs more exploration, it is to somewhat unclear in the current form.

Author Reponses: edited/corrected as commented

Methods: as the majority of participants were from rural regions, it should be notes as a limitation considering that the residential status may be a determinant of perceived barriers.

Author Reponses: edited/corrected as commented

Results: inserting a table including the main themes and sub-themes in the beginning of results is suggested.

Author Reponses: edited/corrected as commented (see supporting information file-4)

Discussion: the first 2 paragraphs of discussion should remove into introduction section. Start your discussion with brief presentation of main results.

Author Reponses: edited/corrected based on the comment 

Discussion: discussion section should be improved by providing some policy recommendation/applications of results to deal with the problem and improve the status.

Author Reponses: edited/corrected based on the comment 

7. PLOS authors have the option to publish the peer review history of their article (what does this mean?). If published, this will include your full peer review and any attached files.

Do you want your identity to be public for this peer review? For information about this choice, including consent withdrawal, please see our Privacy Policy.

Reviewer #1: No

Reviewer #2: Yes: Mohammad Amin Bahrami

---

## [Decision Letter · Decision Letter 1]

2 Jun 2021

PONE-D-21-07870R1

Community Risk perception and Barriers for the Practice of COVID-19 Prevention Measures in Northwest Ethiopia :  A Qualitative Study

PLOS ONE

Dear Dr. Tesfaw,

Thank you for submitting your manuscript to PLOS ONE. After careful consideration, we feel that it has merit but does not fully meet PLOS ONE’s publication criteria as it currently stands. Therefore, we invite you to submit a revised version of the manuscript that addresses the points raised during the review process.

Please address all additional comments from reviewer 1 in your revised submission.

We look forward to receiving your revised manuscript.

Kind regards,

Amy Michelle DeBaets, PhD

Academic Editor

PLOS ONE

Journal Requirements:

Reviewers' comments:

Reviewer's Responses to Questions

**Comments to the Author**

1. If the authors have adequately addressed your comments raised in a previous round of review and you feel that this manuscript is now acceptable for publication, you may indicate that here to bypass the “Comments to the Author” section, enter your conflict of interest statement in the “Confidential to Editor” section, and submit your "Accept" recommendation.

Reviewer #1: (No Response)

Reviewer #2: All comments have been addressed

2. Is the manuscript technically sound, and do the data support the conclusions?

Reviewer #1: Partly

Reviewer #2: Yes

3. Has the statistical analysis been performed appropriately and rigorously? 

Reviewer #1: N/A

Reviewer #2: Yes

4. Have the authors made all data underlying the findings in their manuscript fully available?

Reviewer #1: No

Reviewer #2: Yes

5. Is the manuscript presented in an intelligible fashion and written in standard English?

Reviewer #1: No

Reviewer #2: Yes

6. Review Comments to the Author

Reviewer #1: Thanks for taking the time to update the language and methods that now much more clearly explain the qualitative approach. A number of additional grammatical suggestions are provided below for the abstract, introduction, and methods as well as additional questions/comments. The results section and discussion can still use updating to the language to clarify meaning. I make a few suggestions below for the results section specifically. At the moment the discussion mostly serves to resummarize the results rather than place the findings in context of the ongoing pandemic. For example, can you more specifically state how these findings relate to other studies in Ethiopia, Africa, or other lower income countries? Coming from a public health group could you state a little more about how these findings could be used? Have other areas in the country/continent done a better job in implementing educational campaigns? Can you draw from other campaigns that have done a good job in communicating information to rural areas that may serve COVID-19? Could these findings drive approaches to the vaccine campaign once supplies become available? For example, how might religious beliefs impact the community from taking the vaccine?

Specific comments

Abstract

Methods: Community key informants and health extension workers were selected purposely for in-depth interviews and focus group discussion to [should this be “on” instead of “to”] maintaining WHO recommendations for social distancing and use of appropriate personal protective equipment [delete the 's.]

Specifically state the sample size number in the abstract.

FGD has not been defined in the abstract, this acronym is also not needed and could just state focus group discussions throughout the paper.

Results: Three main themes and five categories were emerged from… and 2 religious leaders were participated...[delete was] conducted among purposely selected community members. The age of the participants [delete was] ranged from 24 to 70 years…

For the last two sentences I would rearrange them so that it states- The major identified barriers for practicing COVID-19 prevention measures were the presence of strong cultural and religious practices, perceiving that the disease does not affect the young, misinformation about the disease, and “the” lack of trust on the prevention measures.

Conclusions: Socio-cultural, religious, and economics related barriers were identified from the participant’s narratives for the practice of COVID-19 prevention measures in south Gondar Zone. Our findings suggest the need to strengthen [delete “ing”] community

Background

The world is closely watching the outbreak of “the” respiratory illness…

The world is now suffering from this global pandemic which [has killed instead of “ kills”] millions

I would suggest the following change for this thought given the widespread availability of vaccines in parts of the world-There are not any proven pharmacological treatments [delete “and vaccines” … “until now”] although a number of trials are [consider adding “currently underway by scientist across” and delete “tested worldwide by several scientists in”] the globe. [Consider adding… “While several effective vaccinations are now available and countries such as the USA and Israel have now vaccinated >50% of adults…something about availability in Ethiopia/Africa continent”]. However, there [delete “are” and add “continue to be” scientifically accepted and recommended prevention measures against COVID-19 infection such as social distancing, frequent hand washing, use of face mask, and proper use of alcohol-based hand rubs or sanitizers (6-8).

With neither treatment nor vaccines [consider adding “readily available”], and [consider deleting “without preexisting immunity” because no one had preexisting immunity to this disease].

For “the effect might be devastating” perhaps you mean, “the virus continues to threaten the country” because of the multiple health challenges the continent [do you mean the country?...if you mean continent you need to specifically state Africa]…[at the end of this paragraph you could add something about the recent spike in India as a reminder of the continued risk of this pandemic even when some parts of the world are returning to pre COVID levels of engagement]

To contain the pandemic, the country established a National Ministerial Committee and the government declared [should be a state of emergency and not the state of emergency] on April 8, 2020. [For the following sentence consider this rewording and numbering the elements…On March 16, 2020 the committee released guidelines that included: 1) 14 days quarantine of passengers (travelers?) in isolated centers; 2) Ethiopian Airlines to stop flying to 30 countries;… 9) temporarily ceasing (do you mean closing? ceasing means taking away from the owners) night clubs and bars; regulating market to avoid unethical exploitation of the situation

In this study, healthcare workers and key informants in the community including local leaders [were not instead of will be] interviewed

The following sentence is unclear/can you update what you are trying to say here-This is helpful from the view of the following perspectives. Healthcare workers the frontline in battling the spread infection as a whole (12) . Secondly, community leaders are also the right of the country in prevention the COVID-19 infection (13).

[Can remove the following sentence because it repeats points noted in the prior section, if desired to keep could say something like.. “The basic protective measures against COVID-19 including distancing, personal protective equipment, and handwashing as noted above are poorly practiced in the community of …] Along with the above scientific explanations, controlling the spread of infection is essential to preventing outbreaks of COVID-19 (14).

Consider this rewording-Basic protective measures against the new coronavirus including stay at home, frequently washing hands with soap and water using alcohol-based hand rub or wash…Thus, the implementation of these practices to reduce the spread of COVID-19 would be challenging in Ethiopia…

RECOMMENDED [RECOMMENDED has not been previously defined or presented this way] prevention measures among community key informants and other community member’s [delete “perspective”] in South Gondar Zone, North west [Northwest?] Ethiopia.

Methods

Study area and period

Can you define Woredas?

This study was conducted on purposely selecting [do you mean selected?, but “purposely selecting could be removed either way”]

Currently, the zone is hardly working…[this statement is too subjective, can it be made more objective by stating how many beds the isolation center/ quarantine center/treatment center is able to accommodate. Similarly, later in the paragraph the 150 cases doesn’t seem like much, but could have more weight if clear that the area would not be able to handle a surge of cases].

Participants of the study and sampling procedures

…from the community [delete “were”] participated [in instead of for] Focus Group Discussions.

Data collection tools and procedures

[Can remove this sentence as it is stated above]-The interview process was continued until the data reach saturation, which defined as the point when answers no longer provide new or additional information on the research question and when recurrent patterns became apparent in the participants descriptions.

Some additional “were/was” before “participated/enhanced” that need to be removed

Results

Table 1-Need to keep the use of capital letters consistent

For quotes can you specifically state which IDI participant provided the quote, so which career were they coming from

The non-quoted paragraphs of “Sociodemographic and economic related barriers” section were confusing, please reword.

“Cultural religious related barriers”

What does this mean? “…did not affect those who had a strong religious practice”

As the participants said, almost all participants [delete “were”] trusted on the use of traditional treatments [delete “which will be”] prepared from different herbal plants. [Delete “As the” participants described, the disease [as the instead of “is like”] common cold and [believed it could] be healed by common traditional treatment modalities of the common cold. When [delete “they were”, “also”, “they said as they will use these traditional treatments prepared by the local traditional healers from different plant leaves and roots and to use holy water.” [the last part is redundant]

Reviewer #2: thanks to authors, all of my comments have been addressed. the manuscript can e accepted in current version.

7. PLOS authors have the option to publish the peer review history of their article (what does this mean?). If published, this will include your full peer review and any attached files.

Reviewer #1: No

Reviewer #2: **Yes: **Assoc. Prof. Dr. Mohammad Amin Bahrami

---

## [Author Response · Author response to Decision Letter 1]

12 Jun 2021

June 8, 2021

 Dear, Editor,

Greetings!

Thank you very much for all the comments provided regarding our manuscript entitled “Community Risk perception and Barriers for the Practice of COVID-19 Prevention Measures in Northwest Ethiopia: A Qualitative Study” which are fully accepted and included in the revised version. I have accordingly made necessary revisions on the paper following the comments and suggestions provided from the reviewer 1. For your kind consideration, please find a point by point response to the comments in the next page of this letter and a submitted new revised version of the manuscript. 

All new changes have been highlighted in dark blue in the main document in order to facilitate review.

I hope that you will find the edits as per your expectation and I look forwards to hear from you

soon.

Yours Sincerely,

 Aragaw Tesfaw (MPH, Epidemiology)

Lecturer, Department of Public health 

College of health sciences, Debre Tabor University 

Email: aragetesfa05@gmail.com

Phone: +251921743820

 Point by point responses to editor comments 

Journal Requirements:

 Author Reponses: We revised and checked the references accordingly based on the comment 

Review Comments to the Author

Reviewer #1: Thanks for taking the time to update the language and methods that now much more clearly explain the qualitative approach. A number of additional grammatical suggestions are provided below for the abstract, introduction, and methods as well as additional questions/comments. The results section and discussion can still use updating to the language to clarify meaning. I make a few suggestions below for the results section specifically. At the moment the discussion mostly serves to re-summarize the results rather than place the findings in context of the ongoing pandemic. For example, can you more specifically state how these findings relate to other studies in Ethiopia, Africa, or other lower income countries? Coming from a public health group could you state a little more about how these findings could be used? Have other areas in the country/continent done a better job in implementing educational campaigns? Can you draw from other campaigns that have done a good job in communicating information to rural areas that may serve COVID-19? Could these findings drive approaches to the vaccine campaign once supplies become available? For example, how might religious beliefs impact the community from taking the vaccine?

Author Reponses: we would like to say thank you for the constructive comments forwarded from the authors which are fully acceptable and revised accordingly based on the suggestions and comments. We tried to edit the grammatical errors and language in each section of the document and we now made necessary revisions. We tried to improve also the discussion section by comparing other studies conducted in Africa and Ethiopia as suggested by the reviewer. We tried to mentioned areas in Ethiopia and Africa which done a better job in implementing educational campaigns regarding the prevention of COVID-19. Regarding the findings of our study, we also mentioned in the discussion section about its how our findings will be used. We tried to mention also other campaigns that have done a good job in communicating information to rural areas that may serve COVID-19. As the reviewer mentioned, our study findings can drive approaches to the vaccine campaign once supplies become available. As we mentioned in the main document, religious beliefs will have an impact the community from taking the vaccine, since we are currently facing the challenge even though there are limited vaccines in the country for high risk groups. Still people are reluctant to take the vaccine due to different perceptions. some studies on vaccine acceptance found that the willing ness to accept the vaccine in Ethiopia is low.

Specific comments

Abstract

Methods: Community key informants and health extension workers were selected purposely for in-depth interviews and focus group discussion to [should this be “on” instead of “to”] maintaining WHO recommendations for social distancing and use of appropriate personal protective equipment [delete the 's.]

Author Reponses: corrected based on the comments 

Specifically state the sample size number in the abstract.

Author Reponses: edited/corrected based on the comments

FGD has not been defined in the abstract, this acronym is also not needed and could just state focus group discussions throughout the paper.

Author Reponses: edited/ corrected based on the comments

Results: Three main themes and five categories were emerged from… and 2 religious leaders were participated...[delete was] conducted among purposely selected community members. The age of the participants [delete was] ranged from 24 to 70 years…

Author Reponses: edited/corrected based on the comments 

For the last two sentences I would rearrange them so that it states- The major identified barriers for practicing COVID-19 prevention measures were the presence of strong cultural and religious practices, perceiving that the disease does not affect the young, misinformation about the disease, and “the” lack of trust on the prevention measures.

Author Reponses: edited/corrected based on the comments

Conclusions: Socio-cultural, religious, and economics related barriers were identified from the participant’s narratives for the practice of COVID-19 prevention measures in south Gondar Zone. Our findings suggest the need to strengthen [delete “ing”] community

Background

The world is closely watching the outbreak of “the” respiratory illness…

The world is now suffering from this global pandemic which [has killed instead of “ kills”] millions

Author Reponses: edited/corrected based on the comments

I would suggest the following change for this thought given the widespread availability of vaccines in parts of the world-There are not any proven pharmacological treatments [delete “and vaccines” … “until now”] although a number of trials are [consider adding “currently underway by scientist across” and delete “tested worldwide by several scientists in”] the globe. [Consider adding… “While several effective vaccinations are now available and countries such as the USA and Israel have now vaccinated >50% of adults…something about availability in Ethiopia/Africa continent”]. However, there [delete “are” and add “continue to be” scientifically accepted and recommended prevention measures against COVID-19 infection such as social distancing, frequent hand washing, use of face mask, and proper use of alcohol-based hand rubs or sanitizers (6-8).

Author Reponses: edited/corrected based on the comments

With neither treatment nor vaccines [consider adding “readily available”], and [consider deleting “without preexisting immunity” because no one had preexisting immunity to this disease].

Author Reponses: edited/corrected based on the comments

For “the effect might be devastating” perhaps you mean, “the virus continues to threaten the country” because of the multiple health challenges the continent [do you mean the country?...if you mean continent you need to specifically state Africa]…[at the end of this paragraph you could add something about the recent spike in India as a reminder of the continued risk of this pandemic even when some parts of the world are returning to pre COVID levels of engagement]

To contain the pandemic, the country established a National Ministerial Committee and the government declared [should be a state of emergency and not the state of emergency] on April 8, 2020. [For the following sentence consider this rewording and numbering the elements…On March 16, 2020 the committee released guidelines that included: 1) 14 days quarantine of passengers (travelers?) in isolated centers; 2) Ethiopian Airlines to stop flying to 30 countries;… 9) temporarily ceasing (do you mean closing? ceasing means taking away from the owners) night clubs and bars; regulating market to avoid unethical exploitation of the situation

In this study, healthcare workers and key informants in the community including local leaders [were not instead of will be] interviewed

The following sentence is unclear/can you update what you are trying to say here-This is helpful from the view of the following perspectives. Healthcare workers the frontline in battling the spread infection as a whole (12) . Secondly, community leaders are also the right of the country in prevention the COVID-19 infection (13).

Author Reponses: edited/corrected based on the comment in the main document 

[Can remove the following sentence because it repeats points noted in the prior section, if desired to keep could say something like.. “The basic protective measures against COVID-19 including distancing, personal protective equipment, and handwashing as noted above are poorly practiced in the community of …] Along with the above scientific explanations, controlling the spread of infection is essential to preventing outbreaks of COVID-19 (14).

Author Reponses: edited/corrected

Consider this rewording-Basic protective measures against the new coronavirus including stay at home, frequently washing hands with soap and water using alcohol-based hand rub or wash…Thus, the implementation of these practices to reduce the spread of COVID-19 would be challenging in Ethiopia…

RECOMMENDED [RECOMMENDED has not been previously defined or presented this way] prevention measures among community key informants and other community member’s [delete “perspective”] in South Gondar Zone, North west [Northwest?] Ethiopia.

Author Reponses: edited/corrected in the main document 

Methods

Study area and period

Can you define Woredas?

Author Reponses: Woredas are the lower administrative areas next to zone called also districts according to the Ethiopian government structure. A number of woreda’s form a zone in combination. Kebele →Woreda→ Zone → Region →Federal 

This study was conducted on purposely selecting [do you mean selected?, but “purposely selecting could be removed either way”]

Author Reponses: edited/corrected based on the comment 

Currently, the zone is hardly working…[this statement is too subjective, can it be made more objective by stating how many beds the isolation center/ quarantine center/treatment center is able to accommodate. Similarly, later in the paragraph the 150 cases doesn’t seem like much, but could have more weight if clear that the area would not be able to handle a surge of cases].

Author Reponses: edited/corrected based on the comment

Participants of the study and sampling procedures

…from the community [delete “were”] participated [in instead of for] Focus Group Discussions.

Author Reponses: edited/corrected based on the comment

Data collection tools and procedures

[Can remove this sentence as it is stated above]-The interview process was continued until the data reach saturation, which defined as the point when answers no longer provide new or additional information on the research question and when recurrent patterns became apparent in the participants descriptions.

Some additional “were/was” before “participated/enhanced” that need to be removed

Author Reponses: edited/corrected based on the comment

Results

Table 1-Need to keep the use of capital letters consistent

For quotes can you specifically state which IDI participant provided the quote, so which career were they coming from

Author Reponses: edited/corrected based on the comment

The non-quoted paragraphs of “Sociodemographic and economic related barriers” section were confusing, please reword.

Author Reponses: edited/corrected based on the comment

“Cultural religious related barriers”

What does this mean? “…did not affect those who had a strong religious practice”

Author Reponses: Cultural and religious barriers were the major perceived barriers mentioned from the participants of the study since there are strong religious and cultural practice in the area. Most people trusts religious rules than the modern health care practice. their trusts on the prevention methods is low. 

 As the participants said, almost all participants [delete “were”] trusted on the use of traditional treatments [delete “which will be”] prepared from different herbal plants. [Delete “As the” participants described, the disease [as the instead of “is like”] common cold and [believed it could] be healed by common traditional treatment modalities of the common cold. When [delete “they were”, “also”, “they said as they will use these traditional treatments prepared by the local traditional healers from different plant leaves and roots and to use holy water.” [the last part is redundant]

Author Reponses: we removed /corrected based on the comment

 Reviewer #2: thanks to authors, all of my comments have been addressed. The manuscript can accepted in current version.

Author response: thank you very much for reviewing and giving constructive comments 

---

## [Decision Letter · Decision Letter 2]

17 Aug 2021

PONE-D-21-07870R2

Community Risk perception and Barriers for the Practice of COVID-19 Prevention Measures in Northwest Ethiopia :  A Qualitative Study

PLOS ONE

Dear Dr. Tesfaw,

Thank you for submitting your manuscript to PLOS ONE. After careful consideration, we feel that it has merit but does not fully meet PLOS ONE’s publication criteria as it currently stands. Therefore, we invite you to submit a revised version of the manuscript that addresses the points raised during the review process.

We look forward to receiving your revised manuscript.

Kind regards,

Johnson Chun-Sing Cheung, D.S.W.

Academic Editor

PLOS ONE

Journal Requirements:

Additional Editor Comments (if provided):

**As there are still a few minor points raised by the reviewer, please address to them carefully and submit your revision to us before we can proceed to publication. In order to speed up the process, only a brief rebuttal letter is required.**

Review Comments to the Author:

Reviewer #1: Thank you once more for taking the time to address my prior comments. A number of grammatical errors remain especially in the discussion which makes the paper at times hard to read. Reading items aloud can sometimes help with these issues. Some suggestions below:

In the abstract, methods: “A total of 23 people were participated in the study” were should be removed.

Similarly in the results of the abstract: Were should be removed in the following sentences… “Three main themes and five categories were emerged”… “and 2 religious leaders were participated”…

Abstract, results: … “(7 participants in each round) was”… was should be were

Abstract, conclusions: … removed the “s” from economics

Background: Update the following sentence “There are no any proven pharmacological treatments developed for the disease until now, although a number of trials are currently underway by scientists across the globe. However, COVID-19 vaccines are developed within the shortest period in the history of vaccine production and now become now available and countries such as” to…There are no pharmacological treatments developed for the disease, although a number of trials are currently underway by scientists across the globe. However, COVID-19 vaccines are now available and countries such as…” (The other text is not needed and can be misleading without additional context about the time of vaccine development).

Background: Please update the following sentence… “…preCOvid levels of engagement, the risk of the pandemic is continued and becomes a global challenge (12). According to the June 8, 2021 Worldometer report, currently India and South Africa…” to …pre-COVID levels of engagement, the risk of the pandemic has continued and is a global challenge (12). According to the June 8, 2021 Worldometer report, India and South Africa…

Background: When listing out the guidelines make sure you are using a “;” after the guideline and not a “:” as you are before 3) and 4). Similarly, you are missing some numbers. There should be a “8” before medical professional training, which will require you to renumber afterwards. And “support the regions preparedness…” will be number 12. Consider the following rewording for the rest of that section. In this study, healthcare workers and key informants in the community including local leaders were interviewed either individually or as part of focus groups to share their perceived barriers to COVID-19 prevention measures. This is helpful from the view of the following perspectives. Healthcare workers are the frontline in battling the spread of infection as a whole (16)…[for the last sentence what does “right of the country” mean? Consider alternative wording for clarity].

In the next paragraph be sure to capitalize “South Gondar Zone” for consistency

Methods: For the data analysis procedure section, did you name the open code software?

Methods, ethical approval and consent to participant: Update… “This is because the study is just a qualitative research which did not have any intervention /clinical trial and did not have follow-ups. In addition, some of the participants were illiterate…” to This level of consent was sufficient because the study was not an intervention/clinical trial and did not have follow-ups. In addition, some of the participants were illiterate…

Results: In addition to some grammar errors and reworking of some sentences that could be shortened for clarity, some work could be done to better organize the quotes. For example, temperature was noted both under Socio-demographic and economic related barriers and Cultural and religious related barriers

Discussion: Consider the following updates… As evidence shows, the recommended prevention methods of COVID-19 are effective measures to control the spread of the pandemic (10, 14, 23). However, the practice of such prevention measures is not consistent regionally as documented in Ethiopia despite efforts are made by the government (11, 15, 24). Therefore, we examined the perceived barriers for the practice of prevention measures and community risk perception of COIVID-19 focusing on both rural and urban settings.

Discussion: For reference 25 you note “less than half percent: which would mean 0.5%...I believe you mean to say less than 50 percent. As the abstract notes “The overall rate of information exposure about COVID-19 was 44.9%.”

Discussion: Consider the following rewrite for paragraph 2- The other important recommended and effective prevention measure is stay at home; however, our findings showed that it is a very difficult preventive measure to practice since most people live a hand--to--mouth way of life. Therefore, they need to get their daily consumption through daily work otherwise many will face hunger. As a result, most people cannot stay at home. Further, because Ethiopia is one of the lowest-income countries in the world and has a very young population, many people live together in a clustered environment, approximately 4-8 people in a single household [note, I don’t think it uncommon for 4 people to live in a single household so maybe you mean to additional note in a single room? Or note the size of the household]. This living situation provides a ripe environment for a contagious diseases like COVID-19. [There is also a sentence about negligence in your text but I’m not sure how the negligence part fits with the above text].

Again, there are several other sections within the discussion that could be updated for clarity such the increased interactions because of religious and cultural practices. The latter is an important point, again one could think back to India’s surge in relation to their religious celebrations or the number of mass outbreaks in the United States as a result of church attendance and singing without masks, but the sentences need to be rewritten to make the point more salient.

---

## [Author Response · Author response to Decision Letter 2]

7 Sep 2021

September 06, 2021

 Dear, Editor,

Greetings!

Thank you very much for all the comments provided regarding our manuscript entitled “Community Risk perception and Barriers for the Practice of COVID-19 Prevention Measures in Northwest Ethiopia: A Qualitative Study” which are fully accepted and included in the revised version. I have accordingly made necessary revisions on the paper following the comments and suggestions provided from the reviewer. For your kind consideration, please find a point by point response to the comments in the next page of this letter and a submitted new revised version of the manuscript. 

All new changes have been highlighted in dark blue in the main document in order to facilitate review.

I hope that you will find the edits as per your expectation and I look forwards to hear from you

soon.

Yours Sincerely,

 Aragaw Tesfaw (MPH/Epidemiology)

Assistant Professor, Department of Public health 

College of health sciences, Debre Tabor University 

Email: aragetesfa05@gmail.com

Phone: +251921743820

I. Point by point responses to reviewer comments 

Reviewer #1: Thank you once more for taking the time to address my prior comments. A number of grammatical errors remain especially in the discussion which makes the paper at times hard to read. Reading items aloud can sometimes help with these issues. Some suggestions below:

In the abstract, methods: “A total of 23 people were participated in the study” were should be removed.

Author response: thank you very much for reviewing and giving constructive comments. We removed as commented by the reviewer 

Similarly in the results of the abstract: Were should be removed in the following sentences… “Three main themes and five categories were emerged”… “and 2 religious leaders were participated”…

Abstract, results: … “(7 participants in each round) was”… was should be were

abstract, conclusions: … removed the “s” from economics

Author response: corrected based on the comment in the main document 

Background: Update the following sentence “There are no any proven pharmacological treatments developed for the disease until now, although a number of trials are currently underway by scientists across the globe. However, COVID-19 vaccines are developed within the shortest period in the history of vaccine production and now become now available and countries such as” to…There are no pharmacological treatments developed for the disease, although a number of trials are currently underway by scientists across the globe. However, COVID-19 vaccines are now available and countries such as…” (The other text is not needed and can be misleading without additional context about the time of vaccine development).

Author response: corrected based on the comment 

Background: Please update the following sentence… “…pre-COvid levels of engagement, the risk of the pandemic is continued and becomes a global challenge (12). According to the June 8, 2021 Worldometer report, currently India and South Africa…” to …pre-COVID levels of engagement, the risk of the pandemic has continued and is a global challenge (12). According to the June 8, 2021 Worldometer report, India and South Africa…

Author response: Updated based on the comment in the main document

Background: When listing out the guidelines make sure you are using a “;” after the guideline and not a “:” as you are before 3) and 4). Similarly, you are missing some numbers. There should be a “8” before medical professional training, which will require you to renumber afterwards. And “support the regions preparedness…” will be number 12. Consider the following rewording for the rest of that section. In this study, healthcare workers and key informants in the community including local leaders were interviewed either individually or as part of focus groups to share their perceived barriers to COVID-19 prevention measures. This is helpful from the view of the following perspectives. Healthcare workers are the frontline in battling the spread of infection as a whole (16)… [for the last sentence what does “right of the country” mean? Consider alternative wording for clarity].

Author response: corrected based on the comment in the main document

In the next paragraph be sure to capitalize “South Gondar Zone” for consistency

Author response: corrected/edited based on the comment 

Methods: For the data analysis procedure section, did you name the open code software?

Author response: open code software version 4.02 /corrected based on the comment in the main document

Methods, ethical approval and consent to participant: Update… “This is because the study is just a qualitative research which did not have any intervention /clinical trial and did not have follow-ups. In addition, some of the participants were illiterate…” to This level of consent was sufficient because the study was not an intervention/clinical trial and did not have follow-ups. In addition, some of the participants were illiterate…

Results: In addition to some grammar errors and reworking of some sentences that could be shortened for clarity, some work could be done to better organize the quotes. For example, temperature was noted both under Socio-demographic and economic related barriers and Cultural and religious related barriers. 

Author response: corrected based on the comment in the main document

Discussion: Consider the following updates… As evidence shows, the recommended prevention methods of COVID-19 are effective measures to control the spread of the pandemic (10, 14, 23). However, the practice of such prevention measures is not consistent regionally as documented in Ethiopia despite efforts are made by the government (11, 15, 24). Therefore, we examined the perceived barriers for the practice of prevention measures and community risk perception of COIVID-19 focusing on both rural and urban settings.

Author response: corrected based on the comment in the main document

Discussion: For reference 25 you note “less than half percent: which would mean 0.5%...I believe you mean to say less than 50 percent. As the abstract notes “The overall rate of information exposure about COVID-19 was 44.9%.”

Author response: thank you! we corrected based on the comment in the main document

Discussion: Consider the following rewrite for paragraph 2- The other important recommended and effective prevention measure is stay at home; however, our findings showed that it is a very difficult preventive measure to practice since most people live a hand--to--mouth way of life. Therefore, they need to get their daily consumption through daily work otherwise many will face hunger. As a result, most people cannot stay at home. Further, because Ethiopia is one of the lowest-income countries in the world and has a very young population, many people live together in a clustered environment, approximately 4-8 people in a single household [note, I don’t think it uncommon for 4 people to live in a single household so maybe you mean to additional note in a single room? Or note the size of the household]. This living situation provides a ripe environment for a contagious diseases like COVID-19. [There is also a sentence about negligence in your text but I’m not sure how the negligence part fits with the above text].

Again, there are several other sections within the discussion that could be updated for clarity such the increased interactions because of religious and cultural practices. The latter is an important point, again one could think back to India’s surge in relation to their religious celebrations or the number of mass outbreaks in the United States as a result of church attendance and singing without masks, but the sentences need to be rewritten to make the point more salient.

Author response: thank you very much for reviewing and giving constructive comments, we tried to modified the document based on the comments 

---

## [Decision Letter · Decision Letter 3]

14 Sep 2021

Community Risk perception and Barriers for the Practice of COVID-19 Prevention Measures in Northwest Ethiopia :  A Qualitative Study

PONE-D-21-07870R3

Dear Dr. Tesfaw,

We’re pleased to inform you that your manuscript has been judged scientifically suitable for publication and will be formally accepted for publication once it meets all outstanding technical requirements.

Kind regards,

Johnson Chun-Sing Cheung, D.S.W.

Academic Editor

PLOS ONE

---

## [Editor Report · Acceptance letter]

17 Sep 2021

PONE-D-21-07870R3 

Community Risk perception and Barriers for the Practice of COVID-19 Prevention Measures in Northwest Ethiopia:  A Qualitative Study 

Dear Dr. Tesfaw:

I'm pleased to inform you that your manuscript has been deemed suitable for publication in PLOS ONE. Congratulations! Your manuscript is now with our production department. 

Kind regards, 

on behalf of

Dr. Johnson Chun-Sing Cheung 

Academic Editor

PLOS ONE